# ASTIH: A collection of axon and myelin segmentation datasets from multiple histology studies

**Armand Collin**[1,2] 🆔                                        ARMAND.COLLIN@POLYMTL.CA

**Mathieu Boudreau**[1] 🆔                                       MATHIEU.BOUDREAU@POLYMTL.CA

**Julien Cohen-Adad**[1,2] 🆔                                    JULIEN.COHEN-ADAD@POLYMTL.CA

[1] *NeuroPoly Lab, Department of Electrical Engineering, Polytechnique Montreal, Montreal, QC, Canada*

[2] *Mila - Quebec Artificial Intelligence Institute, Montreal, QC, Canada*

**Editors:** Accepted for publication at MIDL 2026

## Abstract

Large-scale analysis of axon and myelin morphometry in nervous tissues is fundamental to neuroscience research, yet manual quantification remains a profound bottleneck, limiting the scale and efficiency of studies. To address this, we introduce the Axon Segmentation Training Initiative for Histology (ASTIH), a publicly accessible resource designed to propel the development and validation of automated histomorphometry tools. ASTIH comprises five meticulously curated datasets, standardized for machine learning applications, featuring over 69,000 manually segmented axon fibers. These datasets exhibit significant diversity, spanning three microscopy modalities (TEM, SEM, bright-field), three species (mouse, rat, rabbit), and three distinct anatomical regions (brain, spinal cord, peripheral nerves) with varying pixel resolutions (from 0.2 to 0.002 $\mu m/px$). All datasets contain detailed annotations with standardized boundary delineation between adjacent fibers, enabling effective use for both semantic and instance segmentation tasks. We also provide thoroughly evaluated baseline segmentation models for every dataset in the collection to facilitate future benchmarking.

**Keywords:** Histology, Axon, Myelin, Open Dataset, Segmentation, Microscopy.

## 1. Introduction

The quantitative morphometric analysis of axon fibers at the microstructural level is a crucial component of neuroscience research. Such analyses are indispensable to investigate pathophysiological mechanisms, as direct histological examination provides the most definitive evidence of structural alterations within the nervous system. For example, quantifying the distribution of myelin thickness across subjects can elucidate processes such as remyelination in experimental models of nerve regeneration (Daeschler et al., 2022; Carrillo-Barberà et al., 2023), or characterize demyelination resulting from either pathologies or genetic manipulations (Iram et al., 2024; Bagheri et al., 2024). This microstructural quantification is also useful to validate non-invasive tissue characterization techniques, particularly those derived from Magnetic Resonance Imaging (MRI), through the co-registration of histological data with MRI scans and subsequent correlation of derived biomarkers (Duval et al., 2019; Alyami et al., 2020).

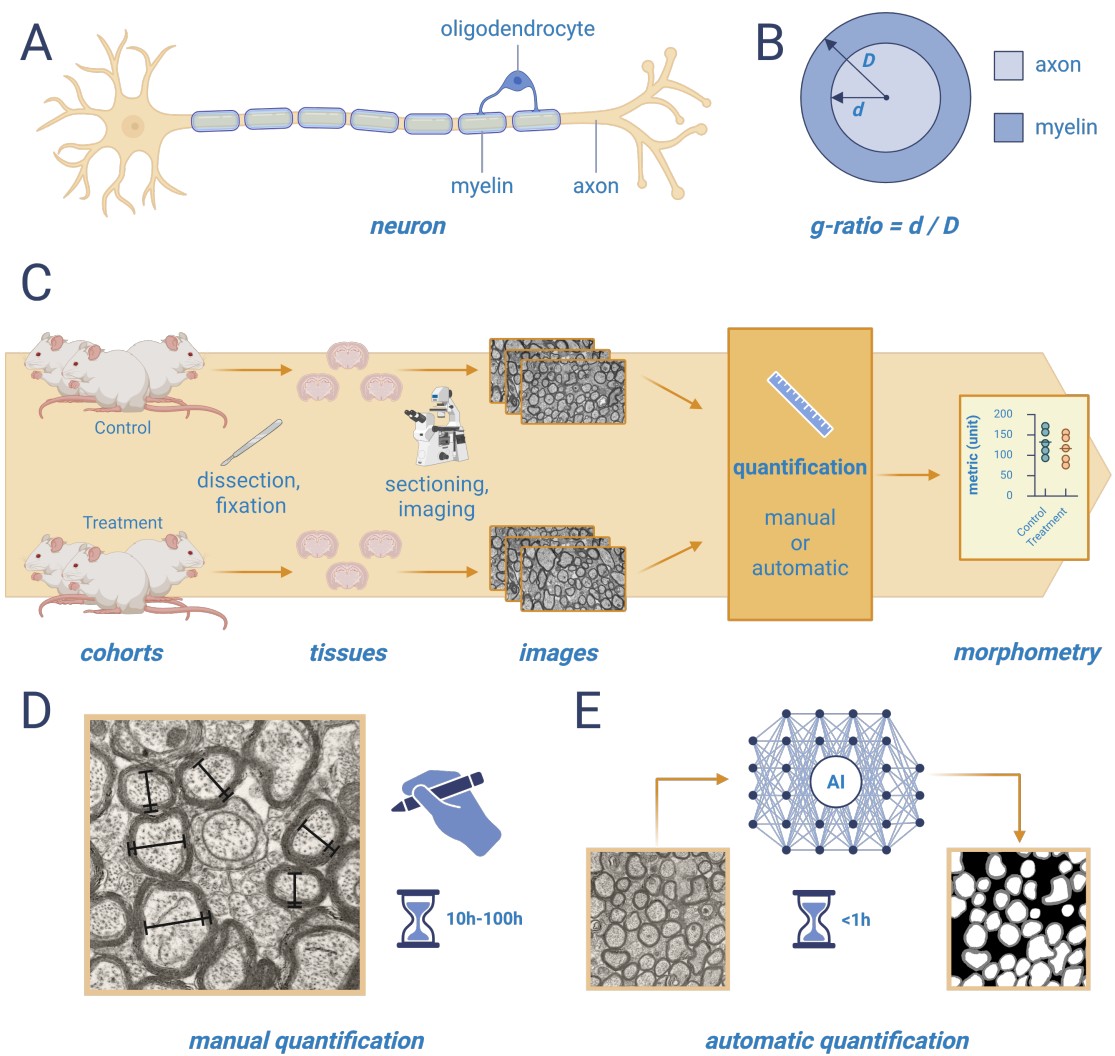

neuron template created by Dr. Veronique Miron from University of Edinburgh

Figure 1: Broad context of ASTIH. (A) Anatomy of a CNS neuron; (B) Illustration of the g-ratio measurement; (C) Typical project involving axon histomorphometry: mice cohorts, upon reaching the desired maturation, are dissected. Tissues are then fixated and sectioned for subsequent microscopy image acquisition and quantification. (D) Manual g-ratio measurements are collected using ImageJ's straight-line tool. (E) Neural networks can automate this process by segmenting the images.

While direct histological examination with manual annotation remains the gold standard for detailed axon morphometry, this approach inherently suffers from several limitations. Primarily, this manual quantification is a time-consuming process, often requiring many hours of focused expert attention, even for medium-size datasets. This method involves

meticulous measurement of numerous features within complex images, a task that significantly restricts the achievable throughput and the scale of studies that can be practically undertaken. Such constraints impose a considerable bottleneck on neuroscientific research, thereby impeding the ability to conduct quantitative investigations at the scale required for high statistical power and comprehensive biological insight. Consequently, there is a pressing and unmet need for robust automated image analysis pipelines capable of accurately segmenting and quantifying axon and myelin morphology across large cohorts. The successful development and widespread adoption of such automated methods are contingent upon the availability of large-scale annotated datasets.

To address this critical gap and facilitate advancements in automated neurohistological analysis, we introduce the Axon Segmentation Training Initiative for Histology (ASTIH), a project that aims to centralize open-source, high-quality datasets curated for training and evaluating segmentation models. ASTIH currently comprises five distinct datasets encompassing different species, anatomical regions and imaging modalities. Our main contributions are:

1. A publicly accessible collection of five histology datasets created to develop and evaluate axon and myelin segmentation models. This collection encompasses unprecedented diversity, featuring over 69,000 manually segmented fibers across three microscopy modalities (TEM, SEM, optical), three species, three regions of the nervous system and various biological conditions.

2. Detailed segmentation masks for every datasets with standardized boundary delineation between adjacent fibers, enabling both semantic and instance segmentation tasks.

3. Pre-trained segmentation models for every dataset, providing a baseline for comparative evaluation of future methods. These models are publicly available and can all be used within the *AxonDeepSeg* open-source software. [1]

All datasets and models can be obtained conveniently on the companion code repository for the ASTIH collection. [2]

## 2. Methods

### 2.1. Datasets

The ASTIH collection contains five curated datasets with over 69,000 manually segmented axon fibers (along with their corresponding myelin sheaths). This resources spans the three most widely used microscopy imaging modalities, two species and three anatomical regions, thereby covering a diverse range of tissue characteristics and imaging resolutions. An overview of the data is presented in Table 1, and previews of the raw images and labels are shown in Fig. 2. Additionally, Table 2 provides basic dataset statistics such as total number of images, number of labels, average number of axon per image, average image size and average foreground-background ratio.

---

1. https://github.com/axondeepseg/axondeepseg
2. https://axondeepseg.github.io/ASTIH

Table 1: Data overview

| Dataset | Region[a] | Species | Resolution[b] | Subjects | Nb. axons | DANDI ID |
|---------|-----------|---------|---------------|----------|-----------|----------|
| TEM1 | B | Mouse | 0.00236 | 20 | 7711 | 001436 |
| TEM2 | B | Mouse | 0.00493 | 10 | 26629 | 001350 |
| SEM1 | SC | Rat | 0.1 | 10 | 7948 | 001442 |
| BF1 | PN | Rat | 0.1 | 8 | 17992 | 001440 |
| BF2 | PN | Rabbit | 0.211 | 14 | 9558 | 001630 |

[a] B: brain, SC: spinal cord, PN: peripheral nerve.
[b] in $\mu$m/px.

### 2.1.1. Data origin

Accumulated over the past decade, the ASTIH datasets originate from diverse biomedical research endeavors. While initially acquired for distinct scientific inquiries, each dataset has been iteratively refined and standardized for deep learning applications. For extensive details about context, tissue preparation, acquisition parameters, and annotation methodology for each constituent dataset, see Appendix A.

Table 2: Dataset statistics.

| Dataset | nb. imgs | nb. GTs | avg axons/img | avg img size | FG/BG ratio |
|---------|----------|---------|---------------|--------------|-------------|
| TEM1 | 158 | 158 | 49 | 3762×2286 | 0.98 |
| TEM2 | 86 | 10 | 437 [a] | 4135×4016 | 4.66 |
| SEM1 | 10 | 10 | 795 | 1225×1064 | 1.84 |
| BF1 | 8 | 8 | 2249 | 5946×5805 | 0.062 |
| BF2 | 14 | 8 | 1195 | 1440×1024 | 0.85 |

[a] Only includes myelinated axons. Average number of unmyelinated axons per image is 2782.

### 2.1.2. Data standardization

We adopted a systematic approach to annotation refinement, quality assurance and data formatting to ensure consistency and interoperability across the entire collection. The primary segmentation targets across all ASTIH datasets are axons and their corresponding myelin sheaths. Although initial annotation methodologies varied according to the original research objectives, a key aspect of our centralized curation process was the consistent implementation of a boundary delineation protocol (see Fig. 3), whereby adjacent axon-myelin units (fibers) were carefully separated. This design choice facilitates robust instance-level evaluation of segmentation algorithms and enables accurate morphometric analysis of individual fibers.

**Annotation guidelines and quality control** The annotation process was divided into two stages. During the first stage, annotators focused on the semantic accuracy of the segmentation. Starting from pseudo-labels generated by semi-automated algorithms, domain experts corrected the masks to ensure accurate myelin thickness for all axons. At this stage,

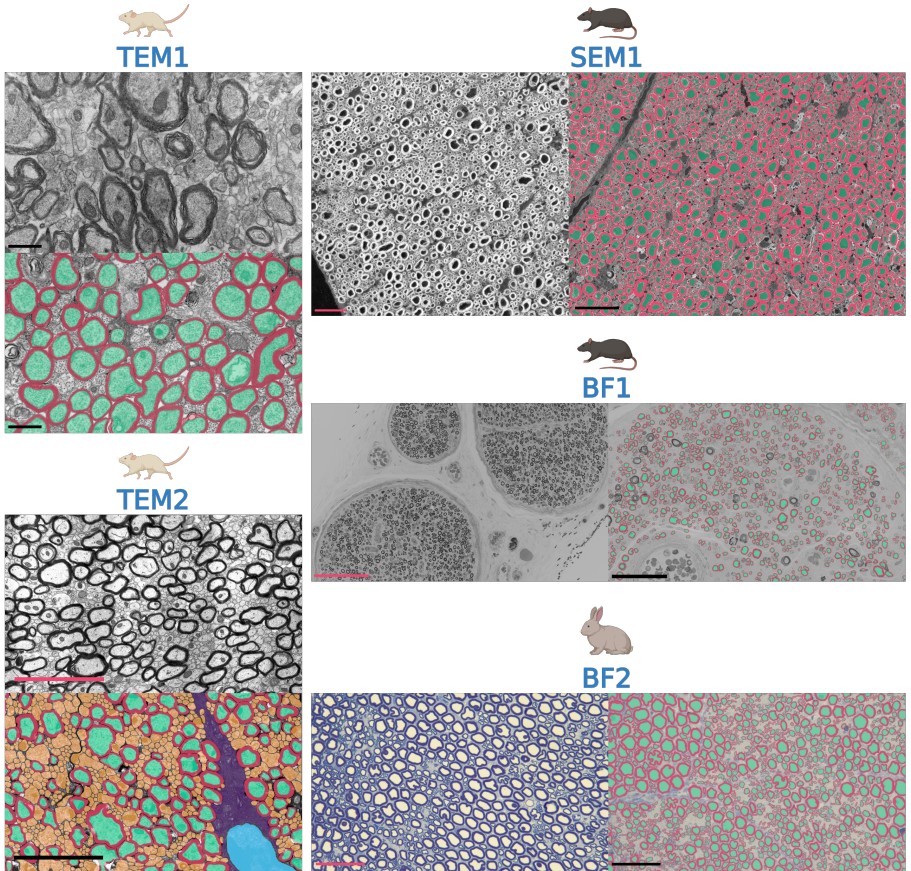

Figure 2: Examples of raw and annotated images for every dataset currently online. Scale-bars: TEM1: 1 um; TEM2: 4 um; SEM1: 30 um; BF1: (left) 100 um, (right) 50 um; BF2: 50 um.

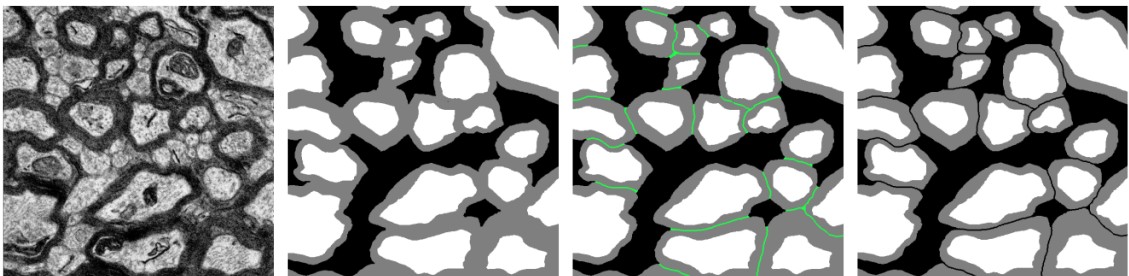

Figure 3: Annotators were instructed to distinctively delineate touching myelin sheaths to preserve instance separation between adjacent fibers.

the myelin sheaths of adjacent, densely packed axons were often merged (see Fig. 3, second panel). Notably, the legacy AxonDeepSeg models (Zaimi et al., 2018) were trained on this earlier version of the data. In the second stage, all masks in the collection underwent a centralized quality control protocol. Following an initial manual separation of adjacent fibers, an automated validation script verified topological constraints across all datasets (e.g., ensuring every myelin region contained exactly one axon region). Any mask violating this rule was flagged and manually corrected, guaranteeing proper instance separation for the entire collection.

**File and directory structure**   Source microscopy images across all datasets are provided as 8-bit grayscale PNG files. The file naming convention and hierarchical directory structure strictly conform to the BIDS-Microscopy data standard (Bourget et al., 2022), an extension of the Brain Imaging Data Structure (BIDS) (Gorgolewski et al., 2016). This standardization framework was originally developed to systematize the organization and metadata of neuroimaging datasets, thereby facilitating automated processing pipelines and inter-operability.

**Annotation masks**   Semantic segmentation masks are supplied as 8-bit PNG files. Consistent file suffixes are employed across the collection for each semantic class: _seg-axon-manual_ and _seg-myelin-manual_ designate the annotated regions for myelinated axons throughout all datasets. As described in A.2, the TEM2 dataset contains additional segmented structures, including unmyelinated axons (_seg-uaxon-manual_), OL nuclei (_seg-nuclei-manual_) and OL processes (_seg-process-manual_). Unmyelinated axons are also resolvable in the TEM1 dataset, but remain unlabeled in this release. In the scanning electron microscopy and bright-field datasets (SEM1, BF1, BF2), the lower resolution, coupled with an incompatible staining mechanism, prevents the distinct visualization of unmyelinated fibers.

**Metadata**   Comprehensive metadata accompany each dataset to describe acquisition parameters, sample preparation protocols and subject-level information. This metadata adheres to the BIDS-Microscopy specification and includes critical information such as pixel resolution, imaging modality, anatomical region and subject characteristics. Consistent with the specification, metadata is organized hierarchically. Image-specific acquisition parameters are stored in JSON sidecars located alongside the image files. Subject-level and sample-level attributes are aggregated in the _participants.tsv_ and _samples.tsv_ 'files at the root of the dataset directory. Global dataset features are defined in the _dataset_description.json_ file. A changelog is also provided to document all modifications made to the datasets over the years.

**Hosting**   Datasets are hosted on the Distributed Archives for Neurophysiology Data Integration (DANDI; https://dandiarchive.org; RRID:SCR_017571), a specialized repository supported by the BRAIN Initiative. The DANDI infrastructure was developed in an effort to encourage open software, explicit licensing and community standards. Each dataset is assigned a persistent Digital Object Identifier (DOI), ensuring long-term accessibility and proper attribution.

### 2.1.3. Data splits for benchmarking

To standardize segmentation model evaluation, we provide official training and testing splits for each dataset. These splits are carefully designed to maintain representativeness while accounting for the inherent characteristics of each dataset. Specifically, for TEM1, SEM1, BF1 and BF2, data partitioning is performed at the subject level, rather than at the image level, to prevent data leakage, as images from the same subject were often acquired consecutively and exhibit similar features. This approach ensures that evaluation metrics accurately reflect the model generalization capabilities to unseen data, thus providing a more realistic assessment of performance in real-world deployment scenarios. For TEM2, as the dataset encompasses all subjects from the original study, the testing set includes images taken from every individual mouse. The accompanying ASTIH codebase facilitates automatic retrieval of testing sets based on the predefined splits specified within each dataset's metadata.

## 2.2. Segmentation models

### 2.2.1. Baselines

**Semantic segmentation**   To establish performance benchmarks and facilitate comparative evaluation of future methods, we implement baseline semantic segmentation models for each dataset in the ASTIH collection. These models are trained using the self-configuring nnU-Net framework (Isensee et al., 2020) (version 2.2), for its competitive performance across diverse biomedical image segmentation tasks. This method allows us to natively train multi-class segmentation models, with separate output channels for axon and myelin classes. This setup is required for automated morphometry of myelin thickness and g-ratio. For all experiments, we employ a consistent training regimen of 1000 epochs using 5-fold cross-validation. The TEM1, SEM1 and BF2 models have 92M parameters, whereas the TEM2 and BF1 models have 126M parameters. Training typically requires between 24 and 48 hours on a single NVIDIA A6000 GPU.

**Instance segmentation**   We additionally train Cellpose models (Stringer et al., 2021) to supplement the benchmark with a native instance segmentation method and demonstrate the dataset's utility for fiber detection tasks. Starting from the cyto3 model weights (Stringer and Pachitariu, 2025), we train Cellpose on the combined axon and myelin masks to segment individual fibers. All models were trained for 500 epochs, except for TEM1, where training was reduced to 300 epochs due to the larger number of images. While this single-class approach does not support myelin morphometry, Cellpose directly optimizes object detection performance at a lower training cost than nnU-Net. The cyto3 model has 6.6M parameters and can be fine-tuned on each dataset in under 10 minutes.

### 2.2.2. Semantic segmentation postprocessing

Since nnU-Net models output semantic masks without distinguishing object instances, we apply the watershed-based post-processing pipeline from AxonDeepSeg (Zaimi et al., 2018) to separate individual fibers (see Fig. 4). This approach leverages the prior that axons are strictly disjoint structures, even when myelin sheaths touch. We compute connected components from the axon mask to identify centroids, which serve as seed points. The exact Euclidian distance transform is applied to the axon mask to create basins for the

watershed algorithm. We apply a marker-controlled watershed algorithm to the negative distance map, constrained by the composite axon-myelin mask. This effectively partitions contiguous regions into discrete axon-myelin units, generating an instance segmentation mask where each unique label corresponds to an individual myelinated fiber, enabling fiber-wise morphometric analysis.

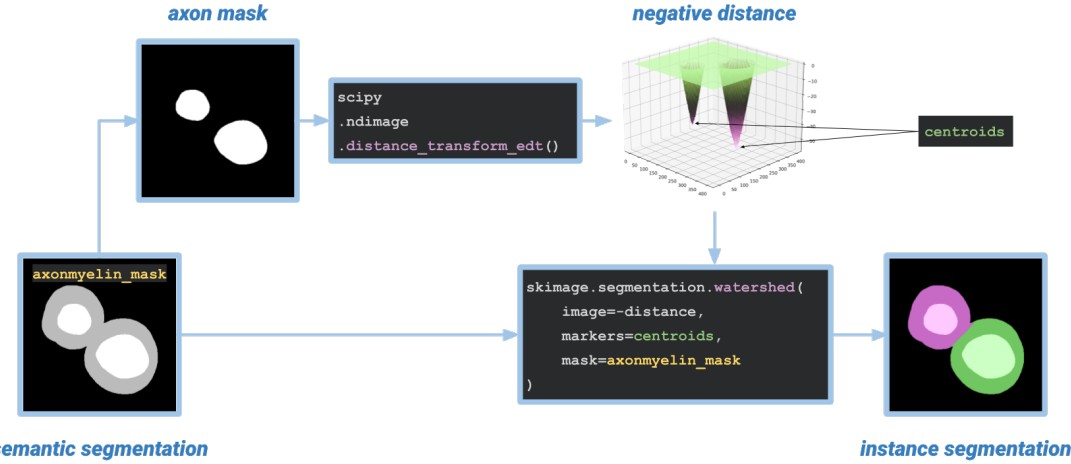

Figure 4: Conversion from semantic to instance segmentation masks.

### 2.2.3. EVALUATION METRICS

To assess pixel-wise segmentation quality, we compute the Dice similarity coefficient and mean Intersection over Union (mIoU) for each semantic class (axon and myelin) on the designated test splits (see Sec. 2.1.3). However, these metrics provide an incomplete assessment of model performance; to quantify the ability to localize individual axons, we additionally compute object-level detection metrics, including precision, recall, and F1-score. To derive these, we apply the postprocessing pipeline described in Sec. 2.2.2 to both prediction and ground-truth masks (although this step is not necessary for Cellpose's native instance segmentation output). Due to the strict boundary delineation enforced during annotation, this yields a perfect instance segmentation for the ground truth. Note that the high density of axons (> 1000 for some images) renders standard matching implementations like `monai.metrics.compute_panoptic_quality` (Cardoso et al., 2022; Kirillov et al., 2019) computationally intractable due to their reliance on dense matrix operations. Instead, we use `stardist.matching` (Schmidt et al., 2018), an algorithm designed for nuclei segmentation that leverages spatial sparsity optimizations to efficiently handle dense instance matching. To decouple detection performance from segmentation precision, we evaluated detection metrics at two IoU thresholds: a permissive one at 0.3, and a more standard value at 0.5.

Table 3: Segmentation quality of nnU-Net models for axon and myelin classes.

| | Axon | | | | Myelin | | | |
| | Dice | | mIoU | | Dice | | mIoU | |
| | mean | std | mean | std | mean | std | mean | std |
|------|-------|---------|-------|--------|-------|---------|-------|--------|
| TEM1 | 0.974 | 0.00542 | 0.950 | 0.0103 | 0.924 | 0.00692 | 0.859 | 0.0118 |
| TEM2 | 0.971 | 0.0217  | 0.945 | 0.0392 | 0.929 | 0.0254  | 0.869 | 0.0432 |
| SEM1 | 0.925 | -       | 0.861 | -      | 0.880 | -       | 0.786 | -      |
| BF1  | 0.856 | 0.0403  | 0.750 | 0.0618 | 0.829 | 0.0203  | 0.708 | 0.0296 |
| BF2  | 0.774 | -       | 0.632 | -      | 0.598 | -       | 0.426 | -      |

## 3. Results

Regarding the nnU-Net evaluation, pixel-wise segmentation quality (Table 3) demonstrates strong baseline performance on TEM datasets (Axon Dice $> 0.97$), while BF models exhibit reduced efficacy (Myelin Dice $\approx 0.60$). BF datasets are acquired at lower magnifications, which limits the amount of details and introduces partial volume effects. This lower performance on optical microscopy data also reflects the difficulty of segmenting deformed axons in regenerating nerves. Fig. 5 illustrates some common segmentation failures of the nnU-Net models. Detection metrics (Table 4) corroborate this trend: while models achieve high detection rates across all modalities at a permissive threshold ($IoU > 0.3$, $F1 > 0.77$), performance on BF data significantly degrades at a stricter threshold ($IoU > 0.5$, $F1 \rightarrow 0.28$). This suggests that while the nnU-Net models correctly identify axon locations in BF images, the precise shape delineation required for strict matching remains challenging.

Table 4 highlights the performance profiles of the two architectures. Cellpose demonstrates exceptional precision ($> 0.97$ in EM) and remarkable stability at stricter IoU thresholds, significantly outperforming nnU-Net on the lower-resolution brightfield (BF) and SEM datasets (e.g., $+0.50$ F1 on BF2 at IoU 0.5). Conversely, nnU-Net achieves higher recall on the high-resolution TEM datasets, indicating it effectively identifies nearly all fibers, though its watershed-based separation is less robust than Cellpose's flow-based masks. We further assessed the robustness of the detection metrics to the matching threshold (see Appendix B).

## 4. Discussion

Prior to ASTIH, publicly available datasets for axon and myelin segmentation in histology images were limited in scope and diversity. Although previous authors have made some of their data public, when taken individually, these resources are insufficient to develop generalizable segmentation models. The gACSON framework (Behanova et al., 2022) and its associated dataset (Sierra et al., 2021) target general brain ultrastructure segmentation in 3D EM volumes. While this resource contains axonal structures, it encompasses a restricted rat cohort ($n = 3$ subjects). It also employs algorithmically-generated annotations rather than manually crafted segmentation masks, potentially limiting its reliability as ground-truth for model training. (Plebani et al., 2022) released a dataset for axon seg-

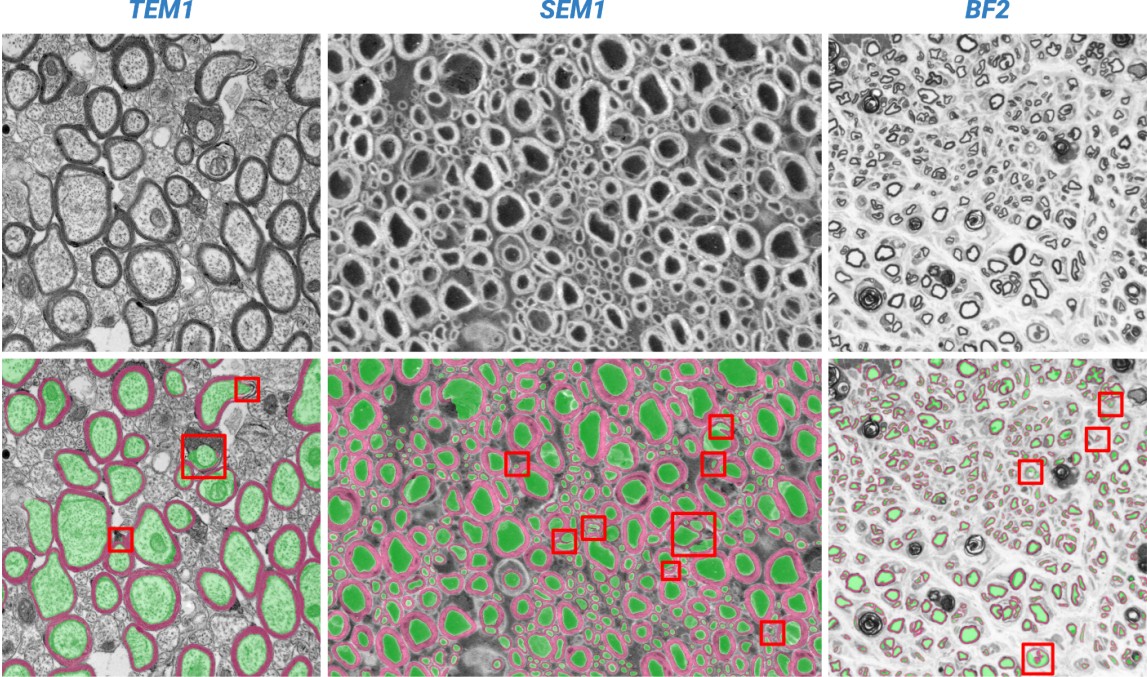

Figure 5: Visual examples of common nnU-Net segmentation failures across modalities. Red boxes highlight spurious predictions.

Table 4: Object detection performance comparison (nnU-Net vs. Cellpose).

| Dataset | nnU-Net | | | Cellpose | | |
| --- | --- | --- | --- | --- | --- | --- |
| | Prec. | Rec. | F1 | Prec. | Rec. | F1 |
| *IoU threshold: 0.3* | | | | | | |
| TEM1 | 0.837 | **0.955** | **0.887** | **0.990** | 0.797 | 0.882 |
| TEM2 | 0.958 | **0.980** | **0.965** | **0.996** | 0.830 | 0.902 |
| SEM1 | 0.860 | **0.873** | 0.866 | **0.971** | 0.794 | **0.873** |
| BF1 | 0.806 | 0.870 | 0.837 | **0.881** | **0.909** | **0.894** |
| BF2 | 0.632 | 0.786 | 0.701 | **0.869** | **0.911** | **0.839** |
| *IoU threshold: 0.5* | | | | | | |
| TEM1 | 0.747 | **0.851** | 0.791 | **0.990** | 0.797 | **0.882** |
| TEM2 | 0.945 | **0.958** | **0.949** | **0.996** | 0.829 | 0.902 |
| SEM1 | 0.701 | 0.712 | 0.707 | **0.968** | **0.791** | **0.871** |
| BF1 | 0.525 | 0.567 | 0.545 | **0.877** | **0.904** | **0.889** |
| BF2 | 0.252 | 0.313 | 0.279 | **0.815** | **0.761** | **0.787** |

mentation in TEM images, with an emphasis on unmyelinated fibers. This resource features high-quality manual annotations covering thousands of fibers, yet it is restricted to one rat cohort ($n = 6$ subjects). This dataset represents one of the few publicly available resource with comprehensive manual annotations. The AimgSeg pipeline (Carrillo-Barberà et al., 2023) also released their dataset consisting of 5 manually annotated corpus callosum images from remyelinating tissues of adult mice ($n = 4$ subjects). The White Matter Microscopy Database (WMMD) (Dyrby et al., 2017) constitutes a broader aggregation of neurohistology images from multiple sources. However, annotations are not consistently available across constituent datasets and the provided segmentation masks were not created manually. Furthermore, the absence of standardized formatting impedes straightforward access and integration into ML pipelines. ASTIH distinguishes itself by addressing these collective limitations through a comprehensive multi-modality and multi-species approach. Key features include the incorporation of diverse pathological conditions, high-quality manual annotations by domain experts, and a standardized data structure. To our knowledge, this initiative is the largest resource to date in terms of diversity ($n = 62$ subjects from 3 different modalities).

Manual g-ratio quantification represents a significant bottleneck, requiring two diameter measurements per axon. In a prior evaluation using the BF1 dataset (Daeschler et al., 2022), manual annotation of $100 \times 100 \ \mu m$ ROIs required 12–29 minutes per ROI, compared to 13–18 seconds for the automated pipeline. Extrapolated to the full cohort of interest (5894 fibers), automation reduced the total analysis time from approximately 13 hours to 6 minutes.

The automated segmentation of axons and myelin sheaths has evolved significantly over the past decade, progressing from traditional image processing approaches to deep learning-based solutions. Early methods relied primarily on intensity thresholding (More et al., 2011; Zhao et al., 2010), contour detection (Richerson et al., 2008; Bégin et al., 2014), morphological operations (More et al., 2011; Richerson et al., 2008; Zhao et al., 2010; Zaimi et al., 2016)] or watershed algorithms (Bégin et al., 2014). These approaches often struggled with variability in tissue preparation, imaging conditions and pathological alterations. More robust traditional methods incorporated active contour models (Bégin et al., 2014; Zaimi et al., 2016). This improved performance but still required extensive parameter tuning and sometimes failed in densely packed fiber regions. The popularization of deep learning approaches, particularly CNNs , ignited a shift towards data-driven methods. Most notably, the U-Net architecture (Ronneberger et al., 2015) was a pivotal work for biomedical image segmentation. This encoder-decoder convolutional network with residual connections is typically trained with heavy data augmentations, and is still a strong baseline for many biomedical image segmentation tasks (Isensee et al., 2020, 2024). Specialized frameworks soon emerged (Moiseev et al., 2018; Plebani et al., 2022), including AxonDeepSeg (Zaimi et al., 2018), which incorporated domain-specific U-Net models. Recently, (Cheng et al., 2023) adopted a different architecture, using a modified version of the pretrained Segment Anything Model (Kirillov et al., 2023) fine-tuned for axon fiber segmentation in EM images.

Parallel to these semantic approaches, generalist instance-segmentation architectures like Cellpose (Stringer et al., 2021) and StarDist (Schmidt et al., 2018) have become standard solutions for cellular segmentation. While semantic models remain advantageous for the specific task of axon morphometry (e.g. g-ratio), these instance-centric frameworks offer

robust alternatives for tasks prioritizing fiber detection and separation in densely packed tissue, where axon density is the metric of interest.

## 5. Conclusion

The Axon Segmentation Training Initiative for Histology (ASTIH) addresses a gap in neuroscience by providing a standardized and centralized resource for automatic axon and myelin segmentation. More than 69,000 fibers were manually segmented in multiple imaging modalities, species, and anatomical regions. ASTIH enables robust development and benchmarking of segmentation algorithms essential for high-throughput neurohistological analysis. While our baseline segmentation models demonstrate strong performance within individual datasets, developing a unified, modality-agnostic segmentation model remains challenging due to significant variations in image appearance and pixel resolution across datasets. Future work should explore transfer learning approaches to bridge these disparities and investigate instance segmentation techniques. We hope this initiative acts as a foundation for advancements in automated axon morphometry. By addressing the manual quantification bottleneck that has traditionally limited research scope, and providing a centralized location for researchers to share and access data, ASTIH aims to provide the tools for an improved understanding of myelination processes in health and disease across the nervous system

Despite ASTIH's comprehensive nature, we acknowledge certain limitations. First, the collection is currently restricted to three species (mouse, rat, and rabbit), which may limit generalizability to other animal models. Second, due to significant resolution disparities (ranging from 0.002 to 0.2 $\mu m/px$), our baseline models remain dataset-specific. Finally, while the inference postprocessing step outputs instance segmentations, the current watershed-based method may struggle in densely packed or pathological tissues.

Our long-term goal is for ASTIH to serve as a centralized community hub for neurohistology data. We invite the community to reach out and collaborate with us to integrate new sources. By actively harmonizing external datasets into our standardized BIDS-Microscopy framework, we plan to unify dispersed resources into a single, reliable engine for automated axon histomorphometry.

## Acknowledgments

We would like to thank Simeon Daeschler, Geetanjali Bendale and Brad Zuchero for agreeing to publish their data; Marie-Hélène Bourget for extending the BIDS standard to microscopy and her contribution to the BF1 dataset; Alexandru Foias, Nick Guenther and Mathieu Guay-Paquet for data management, and Arthur Boschet for training some of the segmentation models. Finally, we would like to thank all contributors to the AxonDeepSeg project, including Aldo Zaimi, Maxime Wabartha, Victor Herman, Pierre-Louis Antonsanti, Christian Perone and Stoyan Asenov.

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

# Appendix A. Detailed Data Origin

## A.1. TEM1: Corpus Callosum of Control and Cuprizone-Intoxicated Mice

The TEM images constituting the TEM1 dataset originated from a study by (Jelescu et al., 2016), which aimed to validate diffusion MRI metrics as potential clinical biomarkers for demyelination. In that work, variations in myelination observed *in vivo* using MRI metrics were correlated with a quantitative EM image analysis of the corresponding tissues. For sample preparation, tissues from the corpus callosum of control and cuprizone-intoxicated mice underwent fixation in a solution containing 2% formaldehyde and 2.5% glutaraldehyde. This was followed by post-fixation in 1% osmium tetroxide overnight. The samples were imaged using a Phillips CM12 transmission electron microscope, with each micrograph capturing an area of approximately $6 \times 9 \ \mu m^2$.

The initial annotation process, as part of the original study, involved segmenting myelin sheaths via a semi-automated procedure reliant on a custom macro in ImageJ; myelinated axon regions were subsequently identified with a connected-component analysis. While unmyelinated axons are visible in these micrographs, the original analysis did not generate segmentation masks for this class, relying instead on manual fiber counts. This dataset was further curated and pre-processed to train deep segmentation models (Zaimi et al., 2018). For its inclusion in ASTIH, a final refinement stage involved a thorough correction of existing segmentation masks, notably to ensure that adjacent touching fibers were separated by a 2 pixel delineation

## A.2. TEM2: Corpus Callosum of Control and SRF Conditional Knockout Mice

The TEM images for the TEM2 dataset originate from recent research (Iram et al., 2024) which identified a pathway regulating the biology of oligodendrocytes (OLs) - the glial cells responsible for myelination in the central nervous system (CNS). This study investigated the mechanistic role of the Serum Response Factor (SRF), a transcription factor found to be indispensable for proper myelin formation in the developing CNS. Accordingly, a key component of their research involved quantitative ultrastructural analysis of myelination in SRF-conditional knockout mice (SRF-cKO). Conditional gene knockout is a technique used to isolate the role of individual genes at specific developmental stages to elucidate gene function. Tissues were fixed with a solution of 2.5% glutaraldehyde, 4% PFA, 13 mM $NaH_2PO_4$, 87 mM $Na_2HPO_4$, and 85.6 mM NaCl for about 10 minutes per mouse. Brains were post-fixed, sectioned to 100-$\mu m$ slices, stained using an osmium-thiocarbohydrazide-osmium method and subsequently underwent high-pressure freezing and freeze substitution (HPF-FS). Ultrathin 90 nm sections were cut using a Leica UC6 ultramicrotome, and imaged using a JEOL JEM-1400 120kV microscope. The published morphometric quantification of the ten subjects (5 SRF-cKO mice, 5 control mice) was made manually in ImageJ

by an expert. This involved direct measurements of axon diameter and myelin thickness using the straight-line tool, and another process focused on counting fibers to determine the percentage of myelinated axons.

The aforementioned manual quantification was a laborious process, illustrating the need for automatic analysis. To this end, the *TEM2* multi-class segmentation dataset was created. For myelinated axons, initial pseudo-labels generated by AxonDeepSeg (Zaimi et al., 2018) were manually corrected, with an emphasis on separation of adjacent fibers by a minimum 2-pixel boundary. Unmyelinated axons were also explicitly segmented, forming a third semantic class. Furthermore, given that the TEM micrographs captured large fields of view (at least 1000 $\mu m^2$), OL cell bodies and their processes—the cytoplasmic extensions responsible for ensheathing axons—were frequently visible. To enhance the dataset's value for investigating OL-axon interactions and to provide a more comprehensive scene context, OL nuclei were manually segmented as a fourth class, and OL processes were delineated as a fifth distinct semantic class. This dataset contains manual annotations for over 26,000 fibers, including 4372 myelinated axons.

### A.3. SEM1: Rat Spinal Cord

The SEM1 dataset, comprising SEM images of rat spinal cord tissues at the cervical level, was developed under experimental protocols approved by the Ethics Committee of the Montreal Cardiology Institute (ICM). A distinctive feature of this dataset is the inherent variability in tissue fixation across different subjects. Specifically, fixative solutions with varying formaldehyde-glutaraldehyde compositions were utilized: most tissues were fixed with 3%-3% concentrations, but 4%-0% and 4%-2% were also used for other rats. The composition of the fixative solution impacts resulting image quality and texture. Following primary fixation, all samples were stained with 2% osmium tetroxide, embedded in epoxy, polished and prepared for imaging. Images were acquired with a Jeol JSM-7600F system, a discontinued SEM microscope. It should also be noted that while standardized procedures were followed, sample preparation and imaging were conducted by different researchers, potentially introducing an additional layer of inter-sample variability in image appearance, thereby presenting a realistic challenge for robust segmentation algorithms.

The segmentation masks associated with SEM1 were initially created to provide training data for the first version of AxonDeepSeg (Zaimi et al., 2018). Preliminary annotations were generated using a semi-automated method (Zaimi et al., 2016), followed by manual correction and formatting suitable for deep learning frameworks. This dataset later served as an exemple for the Microscopy-BIDS standard (Bourget et al., 2022), which advocated for harmonized organization of microscopy data and metadata. For its definitive inclusion in the ASTIH collection, segmentation masks underwent a final refinement stage to ensure that all individual myelinated axons were separated of their neighbors by a 1-pixel boundary. This level of detail, coupled with a marker-controlled watershed algorithm, ensures that each resulting connected component within the mask corresponds to a single, individual fiber, rendering the dataset directly applicable for instance segmentation tasks in addition to semantic segmentation.

### A.4. BF1: Peripheral Rat Nerves at Different Regeneration Stages

The BF1 dataset contains bright-field light microscopy images originating from an experimental study investigating nerve repair mechanisms in rats, as detailed in (Daeschler et al., 2022). The tissue samples were collected from tibial, median and common peroneal nerves in female Sprague-Dawley rats. Notably, to capture various stages of axonal regeneration, samples were harvested from a region 7 to 10 mm distal to the nerve repair site. For histological preparation, tissues were initially fixed in 2.5% glutaraldehyde, followed by staining in 2% osmium tetroxide. Subsequently, samples were embedded in epoxy and sectioned into 1 $\mu$m-thick slices. Sections were imaged using a Zeiss Axiovert 200M microscope equipped with a 63x/1.4 oil objective.

The manual segmentation masks accompanying this dataset were originally prepared to train a bespoke segmentation model for the analyses presented in (Daeschler et al., 2022). The annotation process was performed using GIMP, where annotators carefully traced the inner and outer contours of myelin sheaths with the *free select* tool. This dataset had not been publicly released prior to its inclusion in the ASTIH collection, for which it underwent a final refinement process. This step focused on ensuring all individual myelinated fibers were clearly delineated by a 1 pixel boundary, thereby guaranteeing separability of connected components.

### A.5. BF2: Peripheral rabbit nerve in experimental nerve repair study

The bright-field microscopy images constituting the BF2 dataset originate from a study by (Bendale et al., 2023), which evaluated the efficacy of "Nerve Tape," a sutureless nerve coaptation device utilizing microhooks. In that work, the tibial nerves of New Zealand white rabbits were transected and repaired using either the Nerve Tape device, a commercial conduit, or standard microsutures. Tissue samples were harvested 16 weeks post-injury, distal to the repair site, to assess axonal regeneration. The nerve segments were processed for standard manual histomorphometry, embedded in resin, sectioned, and stained with toluidine blue. The resulting slides were imaged at 40x magnification.

While the original study focused on statistical morphometric analysis (axon counts and g-ratios), these images were also used to create a segmentation model for rabbit tissue. The model trained on the BF1 dataset was used to create preliminary masks, then corrected by an expert. For their inclusion in this collection, the segmentation masks were manually refined to ensure a 1 pixel separation between adjacent fibers, following the standardized ASTIH protocol.

## Appendix B. Sensitivity analysis of IoU threshold

For object-level evaluation, the Intersection over Union (IoU) threshold used in the matching algorithm is a critical parameter. To assess the sensitivity of our metrics to this choice, we computed the F1 score across a range of thresholds (0.3 – 0.99). We selected a test image from the BF1 dataset for this analysis, as it presents a challenging detection scenario characterized by high fiber density (more than 2000 axons). As illustrated in Figure 6, the choice of threshold impacts the two models differently. For nnU-Net, the F1 score decreases linearly across the entire range, whereas for Cellpose, performance remains stable between

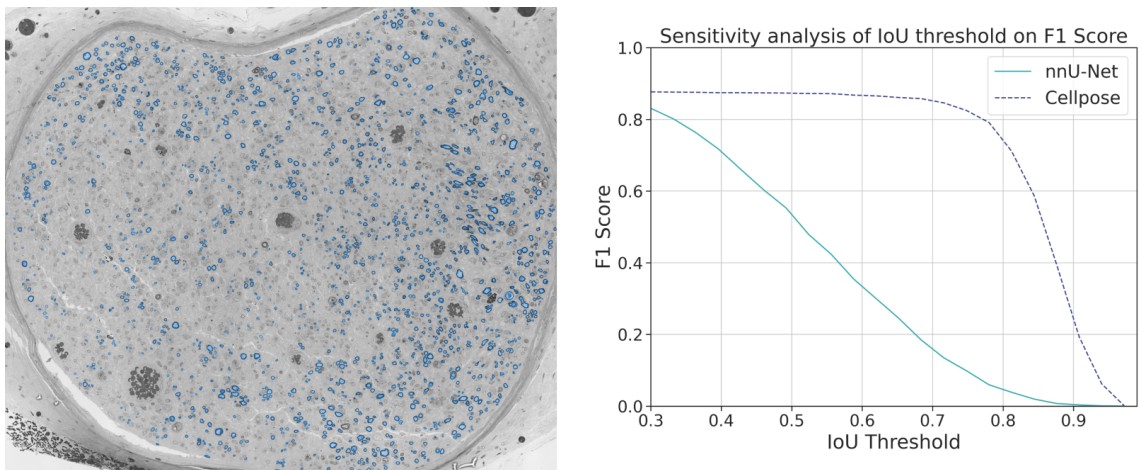

Figure 6: F1 score sensitivity analysis with respect to the IoU threshold, BF1 test image. The GT mask is overlaid in blue on top of the raw image.

0.3 and 0.7, before decreasing sharply. For brevity, we report only the sensitivity analysis for the F1 score, as Precision and Recall exhibited similar trends.

We hypothesize that this difference stems from the underlying segmentation methodologies. The watershed post-processing applied to nnU-Net masks is sensitive to minor boundary erosions or dilations, leading to a gradual, linear reduction in overlap as the strictness increases. In contrast, Cellpose's flow-based reconstruction tends to generate compact, regular shapes that either match the ground truth topology well (high IoU) or miss the instance entirely, resulting in the observed stability.

