# OpenReview forum: "ASTIH: A collection of axon and myelin segmentation datasets from multiple histology studies"
_MIDL.io/2026/Conference — MIDL 2026 Poster_

### Official Review · Reviewer_j4pG · 2026-01-08

**Confidence:** 5
**Preliminary Rating:** 4
**Final Rating:** 5

**Summary:**

The paper contributes to creating a publicly accessible data resource, ASTIH, geared for histomorphometric analysis. In the current stage, it consists of 5 distinct (annotated) data cohorts with significant domain shift (in terms of modality, organism and resolution), vital for developing and validating bioimage analysis tools. What excites me personally is the precise delineation between adjacent structures, making this dataset usable for both semantic and instance segmentation. In addition, the paper trains a model to quantify the importance of annotated images for myelinated axon analysis.

**Strengths:**

1. I like the paper writing and the figures, both are quite self explanatory, communicating the idea very well (with the biological insights in clear sight).
2. Table 1 gives a nice overview. A tiny suggestion would be to add the (average) spatial shape of the images in each dataset. It would give out a flavour of what to expect from the particular cohort.
3. Fig. 2 shows the quality of data generated in the manuscript very well.

Overall, the problem statement and justification for the effort is well motivated!

**Weaknesses:**

I have three major concerns:

1. Comparing the trained model with other models (eg. AxonDeepSeg)
2. Evaluating the pretrained model on downstream task (eg. calculating the g-ratio for myelinated axons)
3. Surfacing back on the data standardization section.

(see the "Questions To Address In The Rebuttal" section below for a detailed outline)

**Detailed Comments:**

Overall, I am confident that the paper is in a great state to be accepted at MIDL as it is, subject to some important questions to be addressed. I would love to see the response from the authors for the comments shared above.
Thanks!

**Justification Of Final Rating:**

I have already been quite happy with the manuscript and open-sourcing motivation right from the very beginning. And the authors did a great job answering all my questions. In particular, this dataset highlights the importance of rich annotations, and discusses details on why they are important for morphometry. Thank you so much for all the efforts put forward by the authors.

**Justification Of The Preliminary Rating:**

I am happy with the manuscript presentation, open-sourcing motivation of the codebase and dataset, and the idea of targeting two critical segmentation tasks with valid downstream analysis tasks. I am open to discuss the response from authors in the coming days.

**Questions To Address In The Rebuttal:**

1. To better support the numbers mentioned in Fig. 1D and Fig. 1E, I suggest the authors add a user study section, either in the main paper or appendix to elaborate on the time taken to quantify the g-ratio measurement. This could be done for a handful of images, but would really reflect and back the demand of the dataset contribution in the paper.
2. Page 4, Section 2.1.2. Data Standardization - How is the metadata stored in the PNG image files? For proper storage and opting for a standard approach in microscopy, wouldn’t it make sense to store images in OME-NGFF (https://www.nature.com/articles/s41592-021-01326-w)? I suggest the authors look into opting to store images and corresponding labels in file formats like ome-tiff (as the simplest choice). This would enable storing associated metadata in the images, enabling multi-modal image analysis in future.
3. Page 6, Section 2.2. Segmentation models - It’s unclear which nnUNet model has been trained (is it v1 (https://www.nature.com/articles/s41592-020-01008-z) / v2 (https://arxiv.org/abs/2404.09556), and in case it’s v2, the size of the residual encoder would be important to report) and the duration of training.
4. Table 2 and Table 3: I am assuming these are numbers evaluated on the ASTIH test-set from the nnUNet model trained on ASTIH train-val dataset. Keeping the focus of the conference in mind, I strongly suggest the authors show comparative results with pretrained AxonDeepSeg models, their previously developed method (https://www.nature.com/articles/s41598-018-22181-4). This would enunciate the importance of pretraining stronger models on high-quality annotated images and justify the strong contribution of such data for DL in bio(medical)imaging applications.
5. How different is the data acquired here compared to the AxonDeepSeg data? (https://www.nature.com/articles/s41598-018-22181-4) Is it the same data as the White Matter Microscopy Database?
My question is in the direction of thinking if it makes sense to combine the two datasets to train a better model? (assuming there’s significant domain shift, but I wonder how good the annotations are for previous data, in case the myelin sheaths aren’t well separated)
Also, same question for AimSeg? (https://doi.org/10.1371/journal.pcbi.1010845)
6. Page 9, Section 4. Discussion -> "To our knowledge, this initiative is the largest resource to date in terms of subject diversity (n = 62)."
Do the authors mean that there are 62 images (and corresponding labels) in the entire ASTIH dataset?
And a follow-up question to this, it would be nice if the authors could mention the total number of images in the entire dataset (I apologize if I missed its mention already).
7. Final comment: I suggest the authors show quantitative assessment of calculating the g-ratio of the myelinated axons using predictions from their best model. This helps the claims of the manuscript come across stronger.

---

> ### Author Response · Authors · 2026-01-24
> **answer to questions 1-3**
>
> We thank the reviewer for their detailed and structured feedback. We will address every point individually:
>
> *1. To better support the numbers mentioned in Fig. 1D and Fig. 1E, I suggest the authors add a user study section, either in the main paper or appendix to elaborate on the time taken to quantify the g-ratio measurement. This could be done for a handful of images, but would really reflect and back the demand of the dataset contribution in the paper.*
>
> We agree that providing more information about analysis time would emphasize the main motivation behind our work. We added the following section to the Discussion section.
> > Manual g-ratio quantification represents a significant bottleneck, requiring two diameter measurements per axon. In a prior evaluation using the BF1 dataset \citep{Daeschler2022}, manual annotation of $100 \times 100$ µm ROIs required 12–29 minutes per ROI, compared to 13–18 seconds for the automated pipeline. Extrapolated to the full cohort of interest (>5800 fibers), automation reduced the total analysis time from approximately 13 hours to just 6 minutes.
>
> *2. Page 4, Section 2.1.2. Data Standardization - How is the metadata stored in the PNG image files? For proper storage and opting for a standard approach in microscopy, wouldn’t it make sense to store images in OME-NGFF (https://www.nature.com/articles/s41592-021-01326-w)? I suggest the authors look into opting to store images and corresponding labels in file formats like ome-tiff (as the simplest choice). This would enable storing associated metadata in the images, enabling multi-modal image analysis in future.*
>
> The metadata is stored in json files according to the microscopy extension of the BIDS specification. This allows us to structure the metadata at the image-level, subject-level, and cohort-level. We added this sentence to clarify this aspect:
> > Consistent with the specification, metadata is organized hierarchically. Image-specific acquisition parameters are stored in JSON sidecars located alongside the image files. Subject-level and sample-level attributes are aggregated in the participants.tsv and samples.tsv files at the root of the dataset directory. Global dataset features are defined in the dataset_description.json file.
>
> We agree that ome-tiff is a strong choice for this type of data and we are interested in this standard for future additions to ASTIH. We currently do not enforce a specific file format for the raw images, so ome-tiff or ome-zarr would be viable alternatives to PNG and TIFF. Note that BIDS-microscopy natively supports these 2 file formats (https://bids-specification.readthedocs.io/en/stable/modality-specific-files/microscopy.html#file-formats)
>
> *3. Page 6, Section 2.2. Segmentation models - It’s unclear which nnUNet model has been trained (is it v1 (https://www.nature.com/articles/s41592-020-01008-z) / v2 (https://arxiv.org/abs/2404.09556), and in case it’s v2, the size of the residual encoder would be important to report) and the duration of training.*
>
> Thank you for pointing out this important detail. Residual encoders were introduced in nnUNet v2.4 (https://arxiv.org/abs/2404.09556 - https://github.com/MIC-DKFZ/nnUNet/releases/tag/v2.4.1). Our models were trained on an earlier version (v2.2). We added this information in the manuscript in section 2.2: Segmentation models. We also added training duration estimates in section 2.2.1.
>
> **Note** We opted not to report precise training times in the final tables. Because our models were trained on shared academic infrastructure with variable GPU load and I/O latency, raw wall-clock training times fluctuated significantly and would not provide a reproducible metric for readers.

---

> > ### Author Response · Authors · 2026-01-24
> > **answer to question 4**
> >
> > *4. Table 2 and Table 3: I am assuming these are numbers evaluated on the ASTIH test-set from the nnUNet model trained on ASTIH train-val dataset. Keeping the focus of the conference in mind, I strongly suggest the authors show comparative results with pretrained AxonDeepSeg models, their previously developed method (https://www.nature.com/articles/s41598-018-22181-4). This would enunciate the importance of pretraining stronger models on high-quality annotated images and justify the strong contribution of such data for DL in bio(medical)imaging applications.*
> >
> > This assumption is correct.
> >
> > Regarding the comparison with legacy models: while we agree that demonstrating improvement over time is valuable, we excluded a direct quantitative comparison with the 2018 AxonDeepSeg models due to two critical constraints:
> > - Confounding variables: Practical software development decisions necessitated re-training the legacy AxonDeepSeg models multiple times since 2018, using different U-Net implementations and training strategies. For example, the 2018 models were trained with the Adam optimizer, whereas nnunetv2 uses SGD with Nesterov momentum and deep supervision. Other confounding variables include differences in data augmentation, activation functions, loss formulation and training length. Because the model architecture and data changed concurrently, it would be hard to isolate the effect of the improved annotation quality on a performance gap.
> > - Data leakage risk: The TEM1 and SEM1 datasets were refined over the years, and the train/test splits have been redefined to create the standardized ASTIH benchmark. Because the exact splits used in 2018 are incompatible with the current data partitions, we cannot guarantee that the 2018 training set has no overlap with the current ASTIH test set. Reporting results with potential data leakage would compromise the rigorous benchmarking standards we aim to establish.

---

> > > ### Author Response · Authors · 2026-01-25
> > > **answer to questions 5-6**
> > >
> > > *5. How different is the data acquired here compared to the AxonDeepSeg data? (https://www.nature.com/articles/s41598-018-22181-4) Is it the same data as the White Matter Microscopy Database? My question is in the direction of thinking if it makes sense to combine the two datasets to train a better model? (assuming there’s significant domain shift, but I wonder how good the annotations are for previous data, in case the myelin sheaths aren’t well separated) Also, same question for AimSeg? (https://doi.org/10.1371/journal.pcbi.1010845)*
> > >
> > > Thank you for these suggestions regarding data integration.
> > > - About **AxonDeepSeg**: ASTIH actually is the cumulative evolution of the data used to train previous AxonDeepSeg models. It includes the previously unpublished original datasets used in our 2018 work (SEM1 and TEM1), but with significant expansions and—crucially—a complete refinement of the annotations to meet the new strict "instance-separation" standard. We added the following subsection to clarify annotation guidelines and situate the original AxonDeepSeg data:
> > > > \paragraph{Annotation guidelines and quality control} The annotation process was divided into two stages. During the first stage, annotators focused on the semantic accuracy of the segmentation. Starting from pseudo-labels generated by semi-automated algorithms, domain experts corrected the masks to ensure accurate myelin thickness for all axons. At this stage, the myelin sheaths of adjacent, densely packed axons were often merged (see Fig. \ref{fig:boundary}, second panel). Notably, the legacy AxonDeepSeg models \citep{Zaimi2018} were trained on this earlier version of the data. In the second stage, all masks in the collection underwent a centralized quality control protocol. Following an initial manual separation of adjacent fibers, an automated validation script verified topological constraints across all datasets (e.g., ensuring every myelin region contained exactly one axon region). Any mask violating this rule was flagged and manually corrected, guaranteeing proper instance separation for the entire collection.
> > > - Regarding **WMMD**: The ASTIH data is distinct from the White Matter Microscopy Database (WMMD). WMMD is a crowd-sourced repository where data is often uploaded without annotations, metadata or standardized quality control. The few annotated samples available there do not currently meet the strict quality and formatting requirements we established for ASTIH (e.g., 1-pixel separation, BIDS structure). Integrating WMMD data would require extensive manual curation and re-annotation, which is outside the current scope, though it remains a promising pool of data for future expansions of the collection.
> > > - Regarding **AimSeg**: We agree that the AimSeg dataset is a high-quality resource with labels that align well with our standards (dual instance/semantic segmentation). Integrating this dataset into ASTIH would perfectly fit our mission of centralizing neurohistology resources. The primary step required is restructuring the data into the BIDS-Microscopy format. We are very interested in this possibility and plan to contact the AimSeg authors to discuss incorporating their data into a future release of the ASTIH collection, as well as Plebani et al. (https://www.nature.com/articles/s41598-022-04854-3).
> > > We added the following sentence to the conclusion:
> > > > Our long-term goal is for ASTIH to serve as a centralized community hub for neurohistology data. We invite the community to reach out and collaborate with us to integrate new sources. By actively harmonizing external datasets into our standardized BIDS-Microscopy framework, we plan to unify dispersed resources into a single, reliable engine for automated axon histomorphometry.
> > >
> > > *6. Page 9, Section 4. Discussion -> "To our knowledge, this initiative is the largest resource to date in terms of subject diversity (n = 62)." Do the authors mean that there are 62 images (and corresponding labels) in the entire ASTIH dataset? And a follow-up question to this, it would be nice if the authors could mention the total number of images in the entire dataset (I apologize if I missed its mention already).*
> > >
> > > “n=62” was meant to communicate 62 subjects. The total number of images is 194. Based on this comment, we think this paragraph in the manuscript was unclear. We edited this section to clarify this aspect.
> > >
> > > Additionally, to address the follow-up question, a table was added with dataset statistics (Table 2: nb of images, average image size, average nb of fibers per image, and approximate foreground-background ratios).

---

> > > > ### Author Response · Authors · 2026-01-25
> > > > **answer to question 7**
> > > >
> > > > *7. Final comment: I suggest the authors show quantitative assessment of calculating the g-ratio of the myelinated axons using predictions from their best model. This helps the claims of the manuscript come across stronger.*
> > > >
> > > > We interpret this comment as a suggestion to compare manual vs automated g-ratio measurements. While we agree that such an evaluation is valuable for the field, we believe it falls outside of the scope of this manuscript. Recent work by Alexander Gow (e.g. https://www.tandfonline.com/doi/pdf/10.1080/17590914.2024.2445624 and https://www.tandfonline.com/doi/pdf/10.1080/17590914.2025.2612034) has highlighted that the standard g-ratio computation can introduce significant artifacts in statistical analysis if not modeled with care. Given these emerging insights, we believe that publishing g-ratio evaluation without a rigorous study of the statistical parameters would be premature and potentially misleading. We are actively investigating these specific methodological challenges in a separate, dedicated follow-up study.

---

> ### Comment · Reviewer_j4pG · 2026-01-30
>
> I thank the authors for putting a lot of effort and consideration to address all the feedback from every reviewer. I am super happy with the rebuttal put forward by the authors.
>
> A few side notes:
>
> > We agree that ome-tiff is a strong choice for this type of data and we are interested in this standard for future additions to ASTIH. We currently do not enforce a specific file format for the raw images, so ome-tiff or ome-zarr would be viable alternatives to PNG and TIFF. Note that BIDS-microscopy natively supports these 2 file formats (https://bids-specification.readthedocs.io/en/stable/modality-specific-files/microscopy.html#file-formats)
>
> Gotcha. I'd say long-term, I would suggest moving to OME-Zarr (super long term for sure), as several open-source repositories (eg. BioImage Archive - https://www.ebi.ac.uk/bioimage-archive/) and many others are actively pivoting towards it to be the true "NGFF" for bioimaging. And your group does have a great stream of research in the presented manuscript's domain. It'll be great to offer the community with standard (meta)data infrastructure (and please take this solely as a piece of suggestion ;))
>
> > Note We opted not to report precise training times in the final tables. Because our models were trained on shared academic infrastructure with variable GPU load and I/O latency, raw wall-clock training times fluctuated significantly and would not provide a reproducible metric for readers.
>
> That's totally fine (and completely understandable).
>
> > Our long-term goal is for ASTIH to serve as a centralized community hub for neurohistology data. We invite the community to reach out and collaborate with us to integrate new sources. By actively harmonizing external datasets into our standardized BIDS-Microscopy framework, we plan to unify dispersed resources into a single, reliable engine for automated axon histomorphometry.
>
> I am personally very happy to hear this! :) (we need more annotated high-quality bioimaging data, according to some common standards). I could pitch you to take a look at MIFA (https://www.nature.com/articles/s41592-025-02835-8) for aligning the trends in microscopy imaging (annotations &) data standards with BIDS! (could be too far fetched of a proposal, but can highly recommend aligning)
>
> > Given these emerging insights, we believe that publishing g-ratio evaluation without a rigorous study of the statistical parameters would be premature and potentially misleading. We are actively investigating these specific methodological challenges in a separate, dedicated follow-up study.
>
> Ah interesting. I'll keep an eye on this. Thanks for following up with detailed references on this.
>
> Once again, thanks to the authors for the manuscript and for presenting such a rich bioimaging dataset paper to MIDL. Looking forward for applications built on ASTIH! :)

---

> > ### Author Response · Authors · 2026-01-31
> >
> > Thank you for the insight about ome-zarr, which we are very interested in. Tangentially, a potential avenue for expansion would be to curate 3D datasets of volume-EM. We are currently exploring this idea, and ome-zarr might just be the perfect choice to store the data.
> >
> > Also, thank you for sharing MIFA. We will take a closer look at these guidelines, as they seem to align perfectly with what our group stands for in terms of dataset reusability.

---

> > > ### Comment · Reviewer_j4pG · 2026-02-01
> > >
> > > > Thank you for the insight about ome-zarr, which we are very interested in. Tangentially, a potential avenue for expansion would be to curate 3D datasets of volume-EM. We are currently exploring this idea, and ome-zarr might just be the perfect choice to store the data.
> > >
> > > Awesome, this definitely sounds like the right direction for storing bioimaging data (I'll keep an eye on your groups' research outcomes ;))
> > >
> > > > Also, thank you for sharing MIFA. We will take a closer look at these guidelines, as they seem to align perfectly with what our group stands for in terms of dataset reusability.
> > >
> > > Sounds great again! :)

---

### Official Review · Reviewer_raw5 · 2026-01-11

**Confidence:** 4
**Preliminary Rating:** 3
**Final Rating:** 4

**Summary:**

This paper presents ASTIH, a curated and publicly available collection of five histology datasets for axon and myelin segmentation, covering multiple species, anatomical regions, and microscopy modalities. The authors describe a standardized annotation protocol, data organization using BIDS-Microscopy, and the release of official train/test splits to support fair benchmarking. Baseline segmentation models are trained using nnU-Net, and both pixel-wise and object-level metrics are reported to characterize performance across datasets. The results highlight strong performance on electron microscopy data and more challenging behavior on bright-field images, emphasizing the need for improved methods in lower-resolution settings. Overall, the work aims to reduce a major bottleneck in neurohistological analysis by enabling scalable and reproducible development of automated segmentation tools.

**Strengths:**

The paper presents a carefully curated and standardized collection of diverse histology datasets that were previously scattered or unavailable. Additionally, the use of a well-established framework (nnU-Net) for baselines is appropriate and provides a strong reference point for future work. Although no new segmentation method is proposed, the dataset itself has high potential value for the neuroscience field.

**Weaknesses:**

Some methodological choices are insufficiently justified or analyzed in depth. For example, annotation quality is assumed to be high, but no quantitative assessment of annotation variability is provided. The evaluation relies heavily on a single baseline architecture and a single postprocessing strategy, which limits insight into how general the reported findings are. While related datasets are discussed, the novelty claims would be stronger with a clearer, quantitative comparison. In addition, the generalization analysis is uneven across datasets, particularly for TEM2 where subject-level separation is not possible.

**Detailed Comments:**

1.Adding visual examples of common segmentation failures would strengthen the results section.
2. A brief subsection describing annotation guidelines and quality control would clarify the dataset curation process.
3. Reporting training time and model size would help readers assess practical usability.
4. Please clarify whether unmyelinated axons are present but unlabeled in datasets other than TEM2.
5. Minor language revisions could improve clarity, particularly in some long discussion sentences.

**Justification Of Final Rating:**

Thanks to the effort the authors have put into this work. The authors' responses and added analyses substantially improved the paper. The rebuttal addressed my concerns, so I am raising my rating to weak accept.

**Justification Of The Preliminary Rating:**

The paper presents a well-curated and standardized dataset that addresses a relevant need in the neuroscience and biomedical image segmentation field. The data quality, annotation effort, and open accessibility are clear strengths. However, this paper is not methodologically novel in terms of algorithms and provides limited analysis beyond baseline experiments using a single framework. Several important aspects, such as annotation consistency, metric sensitivity, and generalization across datasets, are insufficiently explored. With additional analysis and clearer justification, the paper could reach a stronger level.

**Questions To Address In The Rebuttal:**

1. Can the authors provide any quantitative measure of annotation consistency or quality control?
2. Why was nnU-Net selected as the baseline, and do the authors try to utilize other architectures?
3. How sensitive are the object-level metrics to the choice of matching strategy and IoU threshold?
4. Do the authors plan to include cross-dataset or cross-modality generalization benchmarks?
5. How did the authors address the large resolution differences when training unified models?

---

> ### Author Response · Authors · 2026-01-24
>
> We thank the reviewer for recognizing the high potential of the collection for the neuroscience field, and we are grateful for the constructive feedback given.
>
> ## Response to detailed comments
> 1. **Failures figure** A figure with common segmentation failures was added in the results section (see Fig. 5 in revised manuscript).
> 2. **Annotation protocol** As suggested, a subsection was added to clarify the dataset curation process:
> > \paragraph{Annotation guidelines and quality control} The annotation process was divided into two stages. During the first stage, annotators focused on the semantic accuracy of the segmentation. Starting from pseudo-labels generated by semi-automated algorithms, domain experts corrected the masks to ensure accurate myelin thickness for all axons. At this stage, the myelin sheaths of adjacent, densely packed axons were often merged (see Fig. 3, second panel). Notably, the legacy AxonDeepSeg models (Zaimi et al., 2018) were trained on this earlier version of the data. In the second stage, all masks in the collection underwent a centralized quality control protocol. Following an initial manual separation of adjacent fibers, an automated validation script verified topological constraints across all datasets (e.g., ensuring every myelin region contained exactly one axon region). Any mask violating this rule was flagged and manually corrected, guaranteeing proper instance separation for the entire collection.
> 3. **Model size & training time** We added the following sentences in the manuscript to report model size and a ballpark for training time:
> > The TEM1, SEM1 and BF2 models have 92M parameters, whereas the TEM2 and BF1 models have 126M parameters. Training typically requires between 24 and 48 hours on a single NVIDIA A6000 GPU.
> >
> > [...] The cyto3 model has 6.6M parameters and can be fine-tuned on each dataset in under 10 minutes.
>
>  Note: We opted not to report precise training times in the final tables. Because our models were trained on shared academic infrastructure with variable GPU load and I/O latency, raw wall-clock training times fluctuated significantly and would not provide a reproducible metric for readers.
>
> 4. **Unmyelinated axon count** Good catch. We added the following sentence in section 2.1.2:
> > Unmyelinated axons are also resolvable in the TEM1 dataset, but remain unlabeled in this release. In the scanning electron microscopy and bright-field datasets (SEM1, BF1, BF2), the lower resolution, coupled with an incompatible staining mechanism, prevents the distinct visualization of unmyelinated fibers.
>
> 5. **Readability** We thank the reviewer for pointing this out. We broke down the longest sentences in the discussion to improve clarity (see highlighted periods in the revised manuscript).

---

> > ### Author Response · Authors · 2026-01-24
> >
> > ## Response to rebuttal questions
> > 1. **Can the authors provide any quantitative measure of annotation consistency or quality control?**
> > We acknowledge that retrospective inter-rater variability measurements were not feasible due to the scale of the dataset (>69,000 fibers) and its cumulative origin over a decade of research. To ensure object-level consistency across all annotations, we implemented a centralized QC protocol (see added subsection: “Annotation guidelines and quality control”).
> > 2. **Why was nnU-Net selected as the baseline, and do the authors try to utilize other architectures?**
> > We agree that this rationale was not detailed sufficiently in the original manuscript. We selected nnU-Net as our primary baseline because it is a competitive framework for biomedical semantic segmentation. Crucially, our collection focuses on axon morphometry (e.g., measuring myelin thickness and g-ratios), which requires precise, multi-class pixel-wise segmentation of two nested structures: the inner axon and the outer myelin sheath. nnU-Net natively supports this multi-class formulation, making it a natural choice to derive these biological metrics.
> > However, we think that exploring other architectures, particularly those optimized for instance segmentation, is valuable for tasks focused on fiber counting. To address this, we have expanded our benchmark to include fine-tuned Cellpose models.
> > We have updated the 2.2.1 Baselines section to include the rationale for choosing nnU-Net and added a new subsection describing the training and advantages of Cellpose models. A comparative analysis of both architectures’ detection performance was added in the Results section (See **Table 4** for an updated object-detection evaluation; we removed some columns to fit the results of nnunet/cellpose side-by-side).
> > 3. **How sensitive are the object-level metrics to the choice of matching strategy and IoU threshold?**
> > Thank you for raising this important point.
> > Regarding the **matching strategy**, we employed the stardist implementation because it leverages sparse optimization for the Hungarian algorithm. As noted in the paper, standard implementations (such as MONAI's) proved computationally intractable on our images with high axon density, resulting in out-of-memory errors. Since these algorithms aim to solve the same mathematical assignment problem, the choice of implementation primarily dictates computational feasibility rather than the metric values themselves.
> > In contrast, it is evident that the **IoU threshold** has a significant impact on the object-level metrics. To address this, we added a sensitivity analysis in Appendix B. We found that the threshold choice differentiates the two architectures: nnU-Net performance degrades linearly as the threshold increases (due to boundary sensitivity from watershed), whereas Cellpose remains stable up to a threshold of 0.7 before dropping sharply.
> > 4. **Do the authors plan to include cross-dataset or cross-modality generalization benchmarks?**
> > We did not include cross-modality benchmarks in this initial release because the domain gaps between datasets are extreme, making zero-shot generalization effectively impossible. As the reviewer points out in their next point, the datasets differ in resolution by up to two orders of magnitude (e.g., 0.002 µm/px in TEM1 vs. 0.211 µm/px in BF2) and rely on different contrast mechanisms (electron density in EM vs. light absorption in Brightfield). Preliminary internal tests confirmed that models trained on one modality fail completely (Dice close to 0) when applied directly to another modality without adaptation. Therefore, our priority for this paper was to establish strong, reliable intra-domain baselines for each individual dataset. We view cross-modality generalization as a crucial research problem, which ASTIH is now uniquely positioned to facilitate.
> > 5. **How did the authors address the large resolution differences when training unified models?**
> > We clarify that we did not attempt to train a unified model in this work; rather, we provided dataset-specific baselines to establish performance benchmarks for each domain individually. As the reviewer correctly identifies, the large resolution discrepancies constitute the biggest challenge in training a single, modality-agnostic model. This challenge is precisely one of the main motivations behind ASTIH. By providing researchers with standardized data across this wide resolution spectrum, our goal is to drive research into resolution-invariant architectures and domain adaptation techniques. We believe that solving this problem requires a community effort to explore multi-scale or scale-invariant networks and transfer learning strategies, which ASTIH now makes possible.

---

### Official Review · Reviewer_V66s · 2026-01-16

**Confidence:** 4
**Preliminary Rating:** 4
**Final Rating:** 5

**Summary:**

This submission introduces ASTIH, a large and carefully curated collection of axon and myelin segmentation datasets including multiple histology studies, microscopy modalities (TEM, SEM, bright-field), species, and anatomical regions. The dataset has over 69,000 manually segmented fibers with consistent annotation protocols that allow for both semantic and instance-level analysis, and is shared in a standardized, reproducible format useful for machine learning research. The authors provide baseline segmentation models and evaluations across all datasets to establish reference performance. I think ASTIH represents a substantial and valuable resource for neurohistological image analysis and morphometry.

**Strengths:**

The main strength of this work is the dataset contribution itself, which is both large-scale and well-designed. The authors aggregate five heterogeneous histology datasets into a standardized benchmark, covering multiple imaging modalities, species, anatomical regions, and spatial resolutions. The manual annotation effort is substantial, and the consistent delineation of boundaries between adjacent fibers is super valuable for downstream instance-level analysis and morphometry. The dataset is released with appropriate documentation, licensing, and long-term hosting, and they also include baseline models and evaluation protocols, which should be helpful for future users. Overall, this is a high-quality contribution that aligns well with MIDL, even if preclinical image analysis is not the most frequent area of research here.

**Weaknesses:**

The main weakness of the paper is a misalignment between the nature of the task and the modeling and evaluation framework selected by the authors. Axon and myelin segmentation seems to be fundamentally an instance segmentation problem, butt the baselines rely on semantic segmentation models (nnU-Net) combined with post-processing, and emphasize pixel-wise metrics such as Dice (mostly meaningless here). While object-level detection metrics are also reported, they come as a supplement after global overlap analysis, and the modeling choice does not fully exploit instance-aware architectures that are well established in the bioimage analysis community, like Cellpose or StarDist, which are probably better suited to dealing with densely packed, touching fibers without heuristic splitting via watershed. Including at least one such instance segmentation baseline would make a better case for the benchmark section and better align the evaluation with the background of this dataset's most likely users.

A minor limitation is that some basic dataset statistics (e.g., typical image sizes, approximate numbers of instances per image, foreground–background ratios, or size distributions of segmented structures) are not clearly summarized in the paper, which would be useful information for practitioners considering to use this data.

**Detailed Comments:**

- The  annotation protocol is very well described, and the dataset with its release channels appear to be carefully engineered for long-term reuse.

- The baseline evaluations are reasonable for a dataset paper, but could be improved by including at least one native instance segmentation model commonly used in microscopy, and placing more emphasis on object-level metrics as biologists do. The cell pose v2 paper on how to train a cell pose model on your own data has been widely cited and you could say it has become a standard mandatory benchmark: "Cellpose 2.0: how to train your own model", Nature Methods 2022. The authors could check both the model training and the evaluation metrics they use to take inspiration.

- A summary table describing some basic dataset statistics (image size ranges, instance counts, class proportions) would go a long way towards improving usability and clarity.

**Justification Of Final Rating:**

The authors acknowledged and dealt with my (and other reveiwers') concerns, even minor ones. I do not see any reason not to strongly recommend this paper be accepted at MIDL. Congratulations and have a nice conference!

**Justification Of The Preliminary Rating:**

This submission is strong and presents a carefully and executed dataset contribution. The scale, diversity, and standardization of the ASTIH datasets, together with high-quality annotations and reproducible release practices, make this a valuable long-term resource for neurohistological image analysis. The methodological baselines could be better aligned with the instance segmentation nature of the task, but this is a fixable limitation and does not undermine the core contribution. I think this paper meets the standards for acceptance at MIDL and is likely to have sizable impact.

**Questions To Address In The Rebuttal:**

- If time allows, I would appreciate a lot if the authors could include and benchmark an instance segmentation model (e.g., Cellpose, StarDist), given the instance-level nature of the annotations.

- Could the authors provide a brief summary of basic dataset statistics such as image sizes, number of fibers per image, and approximate foreground–background ratios?

---

> ### Author Response · Authors · 2026-01-24
>
> ## Regarding the choice of Baselines (Instance vs. Semantic)
> We thank the reviewer for highlighting the importance of native instance segmentation models like Cellpose and StarDist. We fully agree that for tasks focused on fiber counting and separating dense clusters, these architectures are the gold standard.
>
> We initially selected nnU-Net as our primary baseline because a key application of the ASTIH dataset is axon morphometry (specifically measuring myelin thickness and g-ratio). This requires measuring the distance between the inner and outer boundaries of the myelin sheath. nnU-Net natively handles this multi-class segmentation (axon vs. myelin pixels), whereas standard Cellpose models (trained on a single class) do not natively support this nested topology.
>
> However, to address the reviewer's feedback and provide a comprehensive benchmark:
> 1. We trained Cellpose models on all datasets (using the combined axon+myelin "fiber" class) to serve as a reference for the detection task.
> 2. We added a new section in the manuscript describing the two approaches, guiding users to choose the architecture (Cellpose vs nnunet) that best fits their biological question (counting vs. morphometry).
> 3. We updated the Results to include a side-by-side comparison of detection metrics.
>
> Changes to Manuscript:
> - Section 2.2.1: Added a description of the Cellpose model and training.
> - Section 3 (Results): Modified **Table 4** which now compares F1, Precision, and Recall for both nnU-Net and Cellpose.
> - Discussion: Added the following to the discussion:
>  > Parallel to these semantic approaches, generalist instance-segmentation architectures like Cellpose \citep{cellpose} and StarDist \citep{schmidt2018} have become standard benchmarks for cellular segmentation. While semantic models remain advantageous for the specific task of axon morphometry (e.g. g-ratio), these instance-centric frameworks offer robust alternatives for tasks prioritizing fiber detection and separation in densely packed tissue, where axon density is the metric of interest.
>
> ## Regarding the brief dataset summary
> In light of this suggestion, we added a table with the following information: total number of images, number of labeled images, average number of axons per image, average image size and average foreground-background ratio (see **Table 2** in revised manuscript). We thank the reviewer for helping us improve the readability and clarity of the manuscript.

---

### Author Rebuttal · Authors · 2026-01-25

**Rebuttal:**

Major revisions include the addition of Cellpose instance segmentation benchmarks, a new sensitivity analysis for detection metrics, and a detailed quality control protocol. We have also addressed all reviewer feedback regarding dataset statistics, metadata standardization, and methodological limitations.

Attached, the revised manuscript with changes highlighted.

**Supporting Material:**

/attachment/795ab6198526a2e9f18a02149e4df6a0b3bf3779.pdf

---

### Meta-Review · Area_Chair_6j2K · 2026-02-06

**Recommendation:** Accept (Oral)
**Confidence:** 5

**Metareview:**

The authors were highly responsive to the reviewers’ questions during the rebuttal phase. All reviewers either maintained their strongest recommendation or increased their original scores to support acceptance of this paper.

The major contribution of this work is the creation of a publicly available, large-scale, multi-modality histology dataset with manual annotations to propel machine learning methods for analyzing axon and myelin morphometry in nervous tissues. This resource is of significant value to the MIDL community.

---

### Decision · Program_Chairs · 2026-02-13

Accept (Poster)